# Learning Deep Disentangled Embeddings
# With the F-Statistic Loss

**Karl Ridgeway**
Department of Computer Science
University of Colorado
and Sensory, Inc.
Boulder, Colorado
`karl.ridgeway@colorado.edu`

**Michael C. Mozer**
Department of Computer Science
University of Colorado
Boulder, Colorado
`mozer@colorado.edu`

## Abstract

Deep-embedding methods aim to discover representations of a domain that make explicit the domain's class structure and thereby support few-shot learning. Disentangling methods aim to make explicit compositional or factorial structure. We combine these two active but independent lines of research and propose a new paradigm suitable for both goals. We propose and evaluate a novel loss function based on the $F$ statistic, which describes the separation of two or more distributions. By ensuring that distinct classes are well separated on a subset of embedding dimensions, we obtain embeddings that are useful for few-shot learning. By not requiring separation on all dimensions, we encourage the discovery of disentangled representations. Our embedding method matches or beats state-of-the-art, as evaluated by performance on recall@$k$ and few-shot learning tasks. Our method also obtains performance superior to a variety of alternatives on disentangling, as evaluated by two key properties of a disentangled representation: modularity and explicitness. The goal of our work is to obtain more interpretable, manipulable, and generalizable deep representations of concepts and categories.

The literature on *deep embeddings* (Chopra et al., 2005; Yi et al., 2014a; Schroff et al., 2015; Ustinova & Lempitsky, 2016; Song et al., 2016; Vinyals et al., 2016; Snell et al., 2017) addresses the problem of discovering representations of a domain that make explicit a particular property of the domain instances. We refer to this property as *class* or *category* or *identity*. For example, a set of animal images might be embedded such that animals of the same species lie closer to one another in the embedding space than to animals of a different species. Deep-embedding methods are trained using a *class-aware oracle* which can be queried to indicate whether two instances are of the same or different class. Because this paradigm can handle an arbitrary number of classes, and because the complete set of classes does not have to be specified in advance—as they would be in an ordinary classifier—deep embeddings are useful for *few-shot learning*. A small set of examples of novel classes can be projected into the embedding space, and an unknown instance can be classified by its proximity to the embeddings of the labeled examples.

Similar to deep embeddings, the literature on *disentangling* attempts to discover representations of a set of instances, but rather than making explicit a single property of the instances (class), the goal is to make explicit multiple, independent properties, which we refer to as *factors*. For example, a disentangled representation of animals might include factors indicating its size, length of its ears, and whether it has feet or fins. We will later be more rigorous in defining a disentangled representation, but for now we operate with the informal notion that the factors form a compositional or distributed representation such that with relatively few factors and relatively few values of each factor, the factor values can be recombined to span the set of instances. Disentangling has been explored using either a fully unsupervised procedure (Chen et al., 2016; Higgins et al., 2017) or a semi-supervised

procedure in which a *factor-aware oracle* can be queried to specify a factor along with sets of instances partitioned by factor value (Reed et al., 2014; Kingma et al., 2014; Kulkarni et al., 2015; Karaletsos et al., 2015; Reed et al., 2015).

Despite their overlapping and related goals, surprisingly little effort has been made to connect research in deep embeddings and disentangling. There are two obvious ways to make the connection. First, a factor-aware oracle might be used to train deep embeddings (instead of a class-aware oracle), and hopefully disentangled representations would emerge. Second, a class-aware oracle might be used to train disentangled representations (instead of a factor-aware oracle), and hopefully an embedding suitable for few-shot learning would emerge. We primarily pursue the former approach, but briefly explore the latter as well.

In the next section, we propose a deep-embedding method that is suitable for both few-shot learning of novel classes and for disentangling factors. After describing the algorithm and showing that it obtains state-of-the-art results on the recall@$k$ task that is ordinarily used to evaluate embeddings, we turn to analyzing how well the algorithm disentangles the factors that contribute to class identity. To perform a rigorous evaluation, we put forth formal, quantifiable criteria for disentanglement, and we show that our algorithm outperforms other state-of-the-art deep-embedding methods and disentanglement methods in achieving these criteria.

# 1   Using the $F$ statistic to separate classes

Deep-embedding methods attempt to discover a nonlinear projection such that instances of the same class lie close together in the embedding space and instances of different classes lie far apart. The algorithms mostly have heuristic criteria for determining how close is close and how far is far, and they terminate learning once a solution meets the criterion. The criterion can be specified by a user-adjustable margin parameter (Schroff et al., 2015; Chopra et al., 2005) or by ensuring that every within-class pair is closer together than any of the between-class pairs (Ustinova & Lempitsky, 2016). We propose a method that determines when to terminate using the currency of probability and statistical hypothesis testing. It also aligns dimensions of the embedding space with the underlying generative factors—categorical and semantic features—and thereby facilitates the disentangling of representations.

For expository purposes, consider two classes, $C = \{1, 2\}$, having $n_1$ and $n_2$ instances, which are mapped to a one-dimensional embedding. The embedding coordinate of instance $j$ of class $i$ is denoted $z_{ij}$. The goal of any embedding procedure is to separate the coordinates of the two classes. In our approach, we quantify the separation via the probability that the true class means in the underlying environment, $\mu_1$ and $\mu_2$, are different from one another. Our training goal can thus be formulated as minimizing $\Pr\left(\mu_1 = \mu_2 \mid s(z), n_1, n_2\right)$, where $s(z)$ denotes summary statistics of the labeled embedding points. This posterior is intractable, so instead we operate on the likelihood $\Pr\left(s(z) \mid \mu_1 = \mu_2, n_1, n_2\right)$ as a proxy.

We borrow a particular statistic from analysis of variance (ANOVA) hypothesis testing for equality of means. The statistic is a ratio of between-class variability to within-class variability:

$$s = \tilde{n}\frac{\sum_i n_i(\bar{z}_i - \bar{\bar{z}})^2}{\sum_{i,j}(z_{ij} - \bar{z}_i)^2}$$

where $\bar{z}_i = \langle z_{ij} \rangle$ and $\bar{\bar{z}} = \langle \bar{z}_i \rangle$ are expectations and $\tilde{n} = n_1 + n_2 - 2$. Under the null hypothesis $\mu_1 = \mu_2$ and an additional normality assumption, $z_{ij} \sim \mathcal{N}(\mu, \sigma^2)$, our statistic $s$ is a draw from a Fisher-Snedecor (or $F$) distribution with degrees of freedom 1 and $\tilde{n}$, $S \sim F_{1,\tilde{n}}$. Large $s$ indicate that embeddings from the two different classes are well separated relative to two embeddings from the same class, which is unlikely under $F_{1,\tilde{n}}$. Thus, the CDF of the $F$ distribution offers a measure of the separation between classes:

$$\Pr\left(S < s \mid \mu_1 = \mu_2, \tilde{n}\right) = I\left(\frac{s}{s+\tilde{n}}, \frac{1}{2}, \frac{\tilde{n}}{2}\right) \tag{1}$$

where $I$ is the regularized incomplete beta function, which is differentiable and thus can be incorporated into an objective function for gradient-based training.

Several comments on this approach. First, although it assumes the two classes have equal variance, the likelihood in Equation 1 is fairly robust against inequality of the variances as long as $n_1 \approx n_2$.[1] Second, the $F$ statistic can be computed for an arbitrary number of classes; the generalization of the likelihood in Equation 1 is conditioned on *all* class instances being drawn from the same distribution. Because this likelihood is a very weak indicator of class separation, we restrict our use of the $F$ statistic to class pairs. Third, this approach is based entirely on *statistics* of the training set, whereas every other deep-embedding method of which we are aware uses training criteria that are based on individual instances. For example, the triplet loss (Schroff et al., 2015) attempts to ensure that for specific triplets $\{z_{11}, z_{12}, z_{21}\}$, $z_{11}$ is closer to $z_{12}$ than to $z_{21}$. Objectives based on specific instances will be more susceptible to noise in the data set and may be more prone to overfitting.

## 1.1 From one to many dimensions

Our example in the previous section assumed one-dimensional embeddings. We have explored two extensions of the approach to many-dimensional embeddings. First, if we assume that the Euclidean distances between embedded points are gamma distributed—which turns out to be a good empirical approximation at any stage of training—then we can represent the numerator and denominator in the $F$ statistic as sums of gamma random variables, and a variant of the unidimensional separation measure (Equation 1) can be used to assess separation based on Euclidean distances. Second, we can apply the unidimensional separation measure for multiple dimensions of the many-dimensional embedding space. We adopt the latter approach because—as we explain shortly—it facilitates disentangling.

For a given class pair $(\alpha, \beta)$, we compute

$$\Phi(\alpha, \beta, k) \equiv \Pr\left(S < s \mid \mu_{\alpha k} = \mu_{\beta k}, \; n_\alpha + n_\beta - 2\right)$$

for each dimension $k$ of the embedding space. We select a set, $\boldsymbol{D}_{\alpha,\beta}$, of the $d$ dimensions with largest $\Phi(\alpha, \beta, k)$, i.e., the dimensions that are best separated already. Although it is important to separate classes, they needn't be separated on *all* dimensions because the pair may have semantic similarity or equivalence along some dimensions. The pair is separated if they can be distinguished reliably on a subset of dimensions.

For a training set or a mini-batch with multiple instances of a set of classes $\boldsymbol{C}$, our embedding objective is to maximize the joint probability of separation for all class pairs $(\alpha, \beta)$ on all relevant dimensions, $\boldsymbol{D}_{\alpha,\beta}$. Framed as a loss, we minimize the log probability:

$$\mathcal{L}_F = -\sum_{\{\alpha,\beta\}\in\boldsymbol{C}} \sum_{k\in\boldsymbol{D}_{\alpha,\beta}} \ln \Phi(\alpha, \beta, k)$$

Figure 1.1 shows an illustration of the algorithm's behavior. We sample instances $x_{\alpha 1..N}$, $x_{\beta 1..M}$ from classes $\alpha$ and $\beta$, and choose $N$ and $M$ such that $N \approx M$. The neural net encodes these instances as embeddings $z_{\alpha 1..N}$ and $z_{\beta 1..M}$, with dimensions $k = 1..D$. The variable $\phi(\alpha, \beta)$ indicates the degree of separation for each dimension, where high values (darker) indicate better separation. In this case, dimension 2 has the best separation, with low within-class and high between-class variance. The algorithm maximizes the $d$ largest values of $\phi(\alpha, \beta)$, and sets the loss for all other dimensions equal to zero.

This *F-statistic loss* has four desirable properties. First, the gradient rapidly drops to zero once classes become reliably separated on at least $d$ dimensions, leading to a natural stopping criterion; the degree of separation obtained is related to the number of samples per class. Second, in contrast to other losses, the F-statistic loss is not invariant to rotations in the embedding space; this focus on separating along specific dimensions tends to yield disentangled features when the class structure is factorial or compositional. Third, embeddings obtained are relatively insensitive to the one free parameter, $d$. Fourth, because the loss is expressed in the currency of probability it can readily be combined with additional losses expressed similarly (e.g., a reconstruction loss framed as a likelihood). The following sections demonstrate the advantages of the $F$-statistic loss for classification and for disentangling attributes related to class identity.

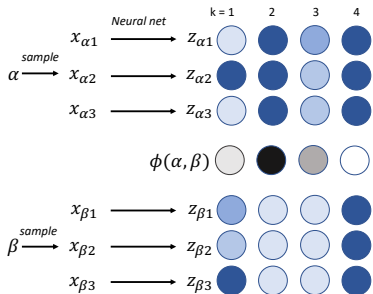

Figure 1: Illustration of the behavior of the $F$-statistic loss for a pair of classes in a minibatch. We sample instances $x_{\alpha 1..N}$, $x_{\beta 1..M}$ from classes $\alpha$ and $\beta$. The neural net encodes these instances as embeddings $z_{\alpha 1..N}$ and $z_{\beta 1..M}$, with dimensions $k = 1..D$. The activations are indicated by the intensity of the blue color. The variable $\phi(\alpha, \beta)$ indicates the degree of separation for each dimension, where high values (darker circle) indicate better separation. In this case, dimension 2 has the best separation, with low within-class and high between-class variance. The algorithm maximizes the $d$ largest values of $\phi(\alpha, \beta)$, and sets the loss for all other dimensions equal to zero.

## 2 Identity classification

In this section, we demonstrate the performance of the $F$-statistic loss compared to state-of-the-art deep-embedding losses on identity classification. The first task involves matching a person from a wide-angle, full-body photograph, taken at various angles and poses. For this task, we evaluate using two datasets—CUHK03 (Li et al., 2014) and Market-1501 (Zheng et al., 2015)—following the methodology of Ustinova & Lempitsky (2016). The second task involves matching a bird from a wide angle photograph; we evaluate performance on the CUB-200-2011 birds dataset (Wah et al., 2011). Five-fold cross validation is performed in every case. The first split is used to tune model hyper-parameters, and we report accuracy on the final four splits. This same procedure was used to evaluate the $F$-statistic loss and four competitors.

### 2.1 Training details

For CUHK03 and Market-1501, we use the Deep Metric Learning (Yi et al., 2014b) architecture, following Ustinova & Lempitsky (2016). For CUB-200-2011, we use an inception v3 (Szegedy et al., 2016) network pretrained on ImageNet, and extract the 2048-dimensional features from the final pooling layer. We treat these features as constants, and optimize a fully connected net, with 1024 hidden ReLU units. For every dataset, we use a 500-dimensional embedding. All nets were trained using the ADAM (Kingma & Ba, 2014) optimizer, with a learning rate of $10^{-4}$ for all losses, except the F-statistic loss, which we found benefitted from a slightly higher learning rate ($2 \times 10^{-4}$). For each split, a validation set was withheld from the training set, and used for early stopping. To construct a mini-batch for training, we randomly select 12 identities, with up to 10 samples of each identity, as in Ustinova & Lempitsky (2016). In addition to the $F$-statistic loss, we evaluated histogram (Ustinova & Lempitsky, 2016), triplet (Schroff et al., 2015), binomial deviance (Yi et al., 2014a), and lifted structured similarity softmax (LSSS) (Song et al., 2016) losses. For the triplet loss, we use all triplets in the minibatch. For the histogram loss and binomial deviance losses, we use all pairs. For the $F$-statistic loss, we use all class pairs. The triplet loss is trained and evaluated using $L_2$ distances. The $F$-statistic loss is evaluated using $L_2$ distances. As in Ustinova & Lempitsky (2016), embeddings obtained discovered by the histogram and binomial-deviance losses are constrained to lie on the unit hypersphere; cosine distance is used for training and evaluation. For the $F$-statistic loss, we determined the best value of $d$, the number of dimensions to separate, using the validation set of the first split. Performance is relatively insensitive to $d$ for $2 < d < 100$. For CUHK03 we chose $d = 70$, for Market-1501 $d = 63$, and for CUB-200 $d = 3$. For the triplet loss we found that a margin of $0.1$ worked well for all datasets. For binomial deviance and LSSS losses, we used the best settings for each dataset as determined in Ustinova & Lempitsky (2016). Code for all models is available at `https://github.com/kridgeway/f-statistic-loss-nips-2018`

### 2.2 Results

Embedding procedures are typically evaluated with either recall@$k$ or with a few-shot learning paradigm. The two evaluations are similar: using held-out classes, $q$ instances of each class are projected to the embedding space (the *references*) and performance is judged by the proximity of a query instance to references in the embedding space. We evaluate with recall@1 or 1-nearest neighbor, which judges the query instance as correctly classified if the closest reference is of the

| Loss | CUHK03 | Market-1501 | CUB-200-2011 |
|---|---|---|---|
| F-Statistic | **90.17% $\pm$ 0.44%** | **84.21% $\pm$ 0.44%** | 55.22% $\pm$ 0.75% |
| Histogram | 86.07% $\pm$ 0.73% | **84.46% $\pm$ 0.23%** | **58.89% $\pm$ 0.89%** |
| Triplet | 81.18% $\pm$ 0.61% | 80.59% $\pm$ 0.64% | 45.09% $\pm$ 0.80% |
| Binomial Deviance | 85.37% $\pm$ 0.45% | **84.12% $\pm$ 0.27%** | **59.05% $\pm$ 0.73%** |
| LSSS | 85.75% $\pm$ 0.62% | 83.46% $\pm$ 0.48% | 54.68% $\pm$ 0.49% |

Figure 2: Recall@1 results for our $F$-statistic loss and four competitors across three data sets. Shown is the percentage correct classification and the standard error of the mean. The best algorithm(s) on a given data set are highlighted.

same class. This is equivalent to a $q$-shot learning evaluation; for our data sets, $q$ ranged from 3 to 10. (For readers familiar with recall@$k$ curves, we note that relative performance of algorithms generally does not vary with $k$, and $k = 1$ shows the largest differences.)

Table 2 reports recall@1 accuracy. Overall, the $F$-statistic loss achieves accuracy comparable to the best of its competitors, histogram and binomial deviance losses. It obtains the best result on CUHK03, ties on Market-1501, and is a tier below the best on CUB-200. In earlier work (Anonymized Citation, 2018), we conducted a battery of empirical tests comparing deep metric learning and few-shot learning methods, and the histogram loss appears to be the most robust. Here, we have demonstrated that our $F$-statistic loss matches this state-of-the-art in terms of producing domain embeddings that cluster instances by class. In the remainder of the paper, we argue that the $F$-statistic loss obtains superior disentangled embeddings.

## 3 Quantifying disentanglement

Disentangling is based on the premise that a set of underlying *factors* are responsible for generating observed instances. The instances are typically high dimensional, redundant, and noisy, and each vector element depends on the value of multiple factors. The goal of a disentangling procedure is to recover the causal factors of an instance in a *code* vector. The term code is synonymous with embedding, but we prefer 'code' in this section to emphasize our focus on disentangling.

The notion of what constitutes an ideal code is somewhat up for debate, with most authors preferring to avoid explicit definitions, and others having conflicting notions (Higgins et al., 2017; Kim & Mnih, 2017). The most explicit and comprehensive definition of disentangling (Eastwood & Williams, 2018) is based on three criteria, which we refer to—using a slight variant of their terminology—as *modularity*, *compactness*, and *explicitness*.[2] In a modular representation, each dimension of the code conveys information about at most one factor. In a compact representation, a given factor is associated with only one or a few code dimensions. In an explicit representation, there is a simple (e.g., linear) mapping from the code to the value of a factor. (See Supplementary Materials for further detail.)

Researchers who have previously attempted to quantify disentangling have considered different subsets of the modularity, compactness, and explicitness criteria. In Eastwood & Williams (2018), all three are included; in Kim & Mnih (2017), modularity and compactness are included, but not explicitness; and in Higgins et al. (2017), modularity is included, but not compactness or explicitness. We argue that modularity and explicitness should be considered as defining features of disentangled representations, but not compactness. Although compactness facilitates interpretation of the representations, it has two significant drawbacks. First, forcing compactness can affect the representation's utility. Consider a factor $\theta \in [0°, 360°]$ that determines the orientation of an object in an image. Encoding the orientation in two dimensions as $(\sin \theta, \cos \theta)$ captures the natural similarity structure of orientations, yet it is not *compact* relative to using $\theta$ as the code. Second, forcing a neural network to discover a minimal (compact) code may lead to local optima in training because the solution space is highly constrained; allowing redundancy in the code enables many equivalent solutions.

In order to evaluate disentangling performance of a deep-embedding procedure, we quantify modularity and explicitness. For modularity, we start by estimating the mutual information between each code dimension and each factor.[3] If code dimension $i$ is ideally modular, it will have high mutual information with a single factor and zero mutual information with all other factors. We use the deviation from this idealized case to compute a modularity score. Given a single code dimension $i$ and a factor $f$, we denote the mutual information between the code and factor by $m_{if}$, $m_{if} \geq 0$. We create a "template" vector $\boldsymbol{t}_i$ of the same size as $\boldsymbol{m}_i$, which represents the best-matching case of ideal modularity for code dimension $i$:

$$t_{if} = \begin{cases} \theta_i & \text{if } f = \arg\max_g(m_{ig}) \\ 0 & \text{otherwise,} \end{cases}$$

where $\theta_i = \max_g(m_{ig})$. The observed deviation from the template is given by

$$\delta_i = \frac{\sum_f (m_{if} - t_{if})^2}{\theta_i^2 (N-1)}, \tag{2}$$

where $N$ is the number of factors. A deviation of 0 indicates that we have achieved perfect modularity and 1 indicates that this dimension has equal mutual information with every factor. Thus, we use $1 - \delta_i$ as a modularity score for code dimension $i$ and the mean of $1 - \delta_i$ over $i$ as the modularity score for the overall code. Note that this expectation does not tell us if each factor is well represented in the code. To ascertain the *coverage* of the code, the explicitness measure is needed.

Under the assumption that factors have discrete values, we can compute an explicitness score for each value of each factor. In an explicit representation, recovering factor values from the code should be possible with a simple classifier. We have experimented with both RBF networks and logistic regression as recovery models, and have found logistic regression, with its implied linear separability, is a more robust procedure. We thus fit a one-versus-rest logistic-regression classifier that takes the entire code as input. We record the ROC area-under-the-curve (AUC) of that classifier. We quantify the explicitness of a code using the mean of $\text{AUC}_{jk}$ over $j$, a factor index, and $k$, an index on values of factor $j$.

In the next section, we use this quantification of modularity and explicitness to evaluate our $F$-statistic loss against other disentangling and deep-embedding methods.

## 4   A weakly supervised approach to disentanglement

Previously proposed disentangling procedures lie at one of two extremes of supervision: either entirely unsupervised (Chen et al., 2016; Higgins et al., 2017), or requiring factor-aware oracles—oracles that name a particular factor and provide sets of instances that either differ on all factors except the named factor (Kulkarni et al., 2015) or are ordered by factor-specific similarity (Karaletsos et al., 2015; Veit et al., 2016). The unsupervised procedures suffer from being underconstrained; the oracle-based procedures require strong supervision.

We propose an oracle-based training procedure with an intermediate degree of supervision, inspired by the deep-embedding literature. We consider an oracle which chooses a factor and a set of instances, then sorts the instances by their similarity on that factor, or into two groups—identical and non-identical. The oracle conveys the similarities but *not* the name of the factor itself. This scenario is like the Sesame Street (children's TV show) game in which a set of objects are presented and one is not like the other, and the child needs to determine along what dimension it differs. Sets of instances segmented in this manner are easy to obtain via crowdsourcing: a worker is given a set of instances and simply told to sort them into two groups by similarity to one another, or to sort them by similarity to a reference. In either case, the sorting dimension is never explicitly specified, and any nontrivial domain will have many dimensions (factors) from which to choose. Our *unnamed-factor oracle* is a generalization of the procedure used for training deep embeddings, where the oracle judges similarity of instances by class label, without reference to the specific class label. Instead, our unnamed-factor oracle operates by choosing a factor randomly and specifying similarity of instances by factor label, without reference to the specific factor.

We explore two datasets in which each instance is tagged with values for several statistically independent factors. Some of the factors are treated as class-related, and some as noise. First, we train on a data set of video game sprites—$60 \times 60$ pixel color images of game characters viewed from various angles and in a variety of poses (Reed et al., 2015). The identity of the game characters is composed of 7 factors—body, arms, hair, gender, armor, greaves, and weapon—each with 2–5 distinct values, leading to 672 total unique identities which can be instantiated in various viewing angles and poses. We also explore the small NORB dataset (LeCun et al., 2004). This dataset is composed of $96 \times 96$ pixel grayscale images of toys in various poses and lighting conditions. There are 5 superordinate categories, each with 10 subordinate categories, a total of 50 types of toys. Each toy is imaged from 9 camera elevations and 18 azimuths, and under 6 lighting conditions. For our experiments, we define factors for toy type, elevation, and azimuth, and we treat lighting condition as a noise variable. For simplicity of evaluation, we partition the values of elevation and azimuth to create binary factors: grouping elevation into low (0 through 4) and high (5 through 8) buckets and azimuth values into right- (0 through 16) and left-(18 through 34) facing buckets, leading to a total of 200 unique identities.

## 4.1 Training Details

For the sprites dataset, we used the encoder architecture of Reed et al. (2015) as well as their embedding dimensionality of 22. For small NORB, we use a convolutional network with 3 convolutional layers and a final fully connected layer with an embedding dimensionality of 20. For the convolutional layers, the filter sizes are ($7 \times 7$, $3 \times 3$, $3 \times 3$), the filter counts are (48, 64, 72), and all use a stride of 2 and ReLU activation. For the $F$-statistic loss, we set the number of training dimensions $d = 2$. Again, all nets were trained using the ADAM optimizer, with the same learning rates as used for the classification datasets.

We construct minibatches in a manner analogous to how we did for deep embeddings with class-based training (Section 3). For factor-based training, we select instances with similarity determined by a single factor to construct a minibatch. For each epoch, we iterate through the factors until we have trained on every instance with respect to every factor. Each minibatch is composed of up to 12 factor-values. For example, a minibatch focusing on the hair color factor of the sprites dataset will include samples of up to 12 hair colors, with multiple instances within each hair color. We train with up to 10 instances per factor-value for triplet and histogram. For the $F$-statistic loss, we found that training with up to 5 instances per factor-value helps avoid underfitting.

For both datasets, we evaluated with five-fold cross validation, using the conjunction of factors to split: the 7 factors for sprites and 3 (toy type, azimuth, and elevation) for norb. For each split, the validation set was used to determine when to stop training, based on mean factor explicitness. The first split was used to tune hyper-parameters, and the test sets of the remaining four splits are used to report results. For these experiments, we compare the $F$-statistic loss to the triplet and histogram losses; other losses using $L_p$ norm or cosine distances should yield similar results. We also compare to the $\beta$-variational auto-encoder, or $\beta$-VAE (Higgins et al., 2017), an unsupervised disentangling method that has been shown to outperform other unsupervised methods such as InfoGAN (Chen et al., 2016). The generator net in the $\beta$-VAE has the same number of layers as the encoder. The number of filters and the size of the receptive field in the generator are mirror values of the encoder, such that the first layer in the encoder has the same number of output filters that the last layer in the generator has as input. For the $\beta$-VAE, training proceeds until the reconstruction likelihood on the held-out validation set stops improving.

## 4.2 Results

Figure 3 shows the modularity and explicitness scores for representations learned on the sprites and small NORB datasets (first and second rows, respectively) using triplet, histogram, and $F$-statistic losses. Modularity scores appear in the first column; for modularity, we report the mean across validation splits and embedding dimensions. Explicitness scores appear in the second column; for explicitness, we report the mean across validation splits and factor-values. (The sprites dataset has 7 factors and 22 total factor-values. The small NORB has a total of 3 factors and 54-factor values.) The $F$-statistic loss achieves the best modularity on both datasets, and the best explicitness on the small NORB dataset. On the Sprites dataset, all of the methods achieve good explicitness.

Figure 4 compares modularity and explicitness of representations for the $F$-statistic and $\beta$-VAE, for various settings of $\beta$. The default setting of $\beta$=1 corresponds to the original VAE (Kingma & Welling,

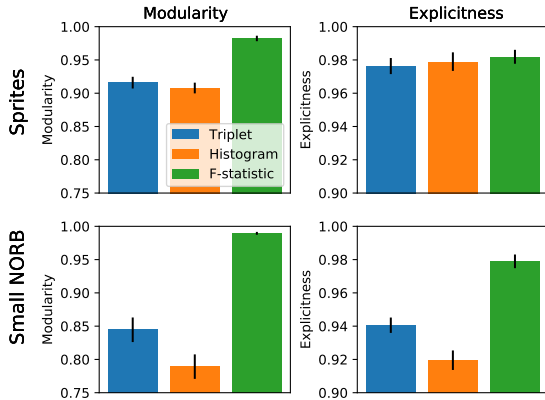

Figure 3: Mean modularity and explicitness scores for the triplet, histogram, and $F$-statistic losses on the small NORB and Sprites datasets. The $F$-statistic loss dominates the other methods in three of the comparisons, and although the $F$-statistic loss has a slight numerical advantage in Sprites explicitness, the advantage is not statistically reliable (comparing histogram to $F$-statistic with a paired $t$-test, $p>.20$). Essentially, all methods are at ceiling in Sprites explicitness. Black bars indicate $\pm$ one standard error of the mean.

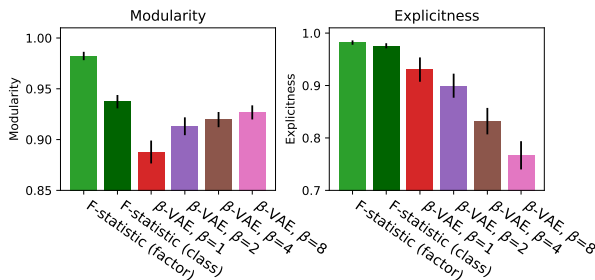

Figure 4: Mean modularity and explicitness scores for the F-statistic loss and $\beta$-VAE on the Sprites dataset. Black bars indicate $\pm$ one standard error of the mean. The light and dark green bars correspond to the $F$-statistic loss trained with an unnamed-factor oracle and a class-aware oracle, respectively. See text for details.

2013). As $\beta$ increases, modularity improves but explicitness worsens. This trade off has not been previously reported and points to a limitation of the method. The first bar of each figure corresponds to the $F$-statistic loss trained with the unnamed-factor oracle, and the second bar corresponds to the $F$-statistic loss trained with a class-aware oracle. The class-aware oracle defines a class as a unique conjunction of the component factors (e.g., for small NORB the conjunction of object identity, azimuth, and elevation). It is thus a weaker form of supervision than the unnamed-factor oracle provides, and is analogous to the type of training performed with deep-embedding procedures, where the oracle indicates whether or not instances match on class without naming the class or its component factors. Both $F$-statistic representations are superior to *all* variants of the $\beta$-VAE. The comparison is not exactly fair because the $\beta$-VAE is unsupervised whereas the $F$-statistic loss is weakly supervised. Nonetheless, the $\beta$-VAE is considered as a critical model for comparison, and we would have been remiss not to do so.

## 5 Discussion and future work

The $F$-statistic loss is motivated by the goal of unifying the deep-embedding and disentangling literatures. We have shown that it achieves state-of-the-art performance in the recall@1 task used to evaluate deep embeddings when trained with a class-aware oracle, and achieves state-of-the-art performance in disentangling when trained with an unnamed-factor oracle. The ultimate goal of research in disentangling is to develop methods that work in a purely unsupervised fashion. The $\beta$-VAE is the leading contender in this regard, but we have shown a troubling trade off obtained with the $\beta$-VAE through our quantification of modularity and explicitness (Figure 4), and we have shown that unsupervised training cannot at present compete with even weakly supervised training (not a surprise to anyone). Another contribution of our work to disentangling is the notion of training with an unnamed-factor oracle or a class-aware oracle; in previous research with supervised disentangling, the stronger factor-aware oracle was used which would indicate a factor name as well as judging similarity in terms of that factor. Our goal is to explore increasingly weaker forms of supervision. We have taken the largest step so far in this regard through our examination of disentangling with a class-aware oracle (Figure 4), which should serve as a reference for others interested in disentangling.

Our current research focuses on methods for adaptively estimating $d$, the hyper-parameter governing the number of dimensions trained on any trial. Presently, $d$ determines the loss behavior for all pairs

of classes, and must be tuned for each data set. Our hope is that we can adaptively estimate $d$ for each pair of identities on the fly.

## 6 Acknowledgements

This research was supported by the National Science Foundation awards EHR-1631428 and SES-1461535.

## Footnotes

[1]For two classes, the $F$-statistic is equivalent to the square of a $t$-statistic. To address the potential issue of unequal variances, we explored replacing the $F$ statistic with the Welch correction for a $t$ statistic, but we found no improvement in model performance, and we prefer formulating the loss in terms of an $F$ statistic due to its greater generality.

[2]We developed our disentangling criteria and terminology in parallel with and independently of Eastwood & Williams (2018). We prefer our nomenclature and also our quantification of the criteria because their quantification requires determination of two hyperparameters (an L1 regularization penalty and a tree depth for a random forest). Nonetheless, it is encouraging that multiple research groups are converging on essentially the same criteria.

[3]In this work, we focus on the case of factors with discrete values and codes with continuous values. We discretize the code by constructing a 20-bin histogram of the code values with equal width bins, and then computing discrete mutual information between the factor-values and the code histogram.

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
