[Supplementary Material · F_Statistic_Loss__NIPS__final_6_supplementary.pdf]

# Supplementary Materials for "Learning Deep Disentangled Embeddings With the F-Statistic Loss"

## A  Desiderata for an Embedding

An embedding is a distributed encoding that captures class (or category or identity) structure via metric properties of the space. To illustrate, the middle panel of Figure 1 shows a projection of instances of varying classes to a 2D space. The projection separates inputs by class and therefore facilitates categorization of unlabeled instances via proximity to the clusters. Such an embedding also allows new classes to be learned with a few labeled examples by projecting these examples into the embedding space. The literature is somewhat splintered between researchers focusing on deep embeddings which are evaluated via few-shot learning (e.g., Ustinova & Lempitsky, 2016; Schroff et al., 2015; Yi et al., 2014) and researchers focusing on few-shot learning who have found deep embeddings to be a useful method (e.g., Snell et al., 2017; Triantafillou et al., 2017).

Figure 1: Alternative two-dimensional embeddings of a domain. Circles represent instances of the domain and color identifies the class. In the leftmost frame, the circles are superimposed.

Figure 1 illustrates a fundamental trade off in formulating an embedding. From left to right frames, the intra-class variability increases and the inter-class structure becomes more conspicuous. In the leftmost panel, the clusters are well separated but the classes are all equally far apart. In the rightmost panel, the clusters are highly overlapping and the blue and orange cluster centers are closer to one another than to the green. Separating clusters is desirable, but so is capturing inter-class similarity. If this similarity is suppressed, then instances of a novel class will not be mapped in a sensible manner—a manner sensitive to input features, underlying factors, and their correspondence. The middle panel reflects a compromise between discarding all variation among instances of the same class and preserving relationships among the classes. With this compromise, embeddings can be used to model hierarchical class structure and can facilitate decomposing the instances according to underlying factors, e.g., separating content and style Tenenbaum & Freeman (2000).

Figure 2: Sixteen different code spaces to represent a set of instances which vary on either two factors (the eight cases on the left) or one factor (the eight cases on the right). Symbol shape and color are used to indicate values of the two factors.

## B   Criteria for Disentangling

In this section, we illustrate the criteria of modularity, compactness, and explicitness using a set of sample codes (Figure 2), in order to compare the criteria against one's intuitions about disentangled representations.

| | Modular | Compact | Explicit |
|---|---|---|---|
| | ✓ | ✓ | ✓ |
| | | | ✓ |
| | ✓ | | ✓ |
| | ✓ | | ✓ |
| | | ✓ | ✓ (for factor indicated by color) |
| | | | ✓ (for factor indicated by color) |
| | ✓ | ✓ | |
| | ✓ | | |
| | ✓ | ✓ | |
| | ✓ | | |
| | ✓ | | |
| | | ✓ | |
| | ✓ | ✓ | |
| | ✓ | | |

Table 1: Modularity, compactness, and explicitness of the sixteen codes in Figure 2. Each code is depicted by an icon that captures the class structure in the embedding space. A check mark indicates that a code satisfies a given disentangling criterion.

In a *modular* representation, each dimension of the code conveys information about at most one factor. Consequently, the code dimensions can be partitioned such that each factor is associated with a single partition and the code for that partition is invariant to the other factors. Figure 2a, which we will depict with the icon ▮, shows a code in which the two factors are modular. The individual points represent a code for a particular instance. The color (red versus magenta) denotes the value of one binary factor, and the symbol (o versus x) denotes the value of a second binary factor. The horizontal and vertical dimensions of the code map to the first and second factors, respectively. In contrast, Figures 2b,e,f (◆, ▬, ◢) present codes in which the two factors are non-modular.

In a *compact* representation, a given factor is associated with only one or a few code dimensions. Figure 2g (▬) shows a code in which a single factor has a compact code. The factor has four distinct values, denoted by the symbols, which are distinguished along the horizontal dimension. In contrast, Figures 2c,d,h (■, ◆, ✎) present codes in which two code dimensions convey information about the single factor.

In an *explicit* representation, the value of a given factor can be precisely determined from the code. The eight lower panels of Figure 2 show noisy versions of the codes in the eight upper panels. Due to the scattering of the points, the code does not permit us to recover the value of every factor for every observation. Thus, a code may fail on the criterion of explicitness because the code-conditional entropy of the factor is nonzero. However, the mutual information between code and factors is only one aspect of explicitness. Compare the codes in Figures 2a (▮) and 2e (▬). For the factor whose values are distinguished by the symbols o and x, we can recover the factor values from code ▮ using a linear separator; however, no linear separator is sufficient to recover the factor values from code ▬. Although the mutual information between the factor and the code is 1 bit for both ▮ and ▬, the factor has more explicit representation in code ▮ than in code ▬ because less computation is required to recover the factor values. The amount of computation required of course depends on computational primitives, but in the disentangling literature, there appears to be an implicit hierarchy of simplicity: an axis-aligned linear discriminant is a simpler operator than a linear discriminant function of all code dimensions, which in turn is simpler than a conjunction of linear discriminant functions, etc. To present another example, one might argue that the codes in Figures 2c,d (■, ◆) are more explicit than those in Figures 2g,h (▬, ✎): for codes ■ and ◆, a boolean predicate on any factor value can be computed by a linear discriminant, whereas for codes ▬ and ✎, a conjunction of linear discriminants is required. However, if recovery is performed by a Gaussian classifier, all four codes are explicit. We have argued that whether a code is explicit or not depends on the way in which it is operated on at subsequent stages of processing. In the context of deep learning, codes are often passed through a neural net with a standard dot-product activation function. Consequently, linear separability is a natural criterion for explicitness with categorical factor values.

Table 1 summarizes the 16 codes in Figure 2 according to whether or not they satisfy the three criteria. For explicitness, we require linear separability. In our examples, the criteria are pretty much satisfied or not, but of course one should specify measures that quantify the degree to which a criterion is satisfied. In our examples, the factors have categorical values—either 2 or 4 values. The modularity and compactness criteria apply directly for continuous-valued factors, but the explicitness criterion requires that we specify a function that recovers the continuous factor value, e.g., a linear function Eastwood & Williams (2018).

Figure 3: Quiver plot of gradients in a two dimensional space for $F$-statistic and triplet losses. Each circle corresponds to an instance, with color indicating class. The arrows represent the gradient direction and magnitude as computed by each loss function.

## C  Effect of Sample Size on Gradients of Loss

This *$F$-statistic loss* gradient rapidly drops to zero once classes become reliably separated, leading to a natural stopping criterion; the degree of separation obtained is related to the number of samples per class. This property is illustrated with synthetic two-dimensional data in the quiver plot in Figure 3, which shows gradients for both the $F$-statistic (with $d = 1$) and triplet losses. For the triplet loss, gradients are averaged over all the possible triplets in the batch. Each box represents a different batch of data. Each column has the same between-class separation on the horizontal axis, with low separation on the left and high separation on the right. Each row has the same number of samples per class. When the classes are reliably separated, with a large between-class distance, the $F$-statistic gradients drop to zero, but the triplet loss will continue pushing the instances apart.

## D  Qualitative Evaluation of Disentanglement

Figure 4 shows a qualitative evaluation of disentanglement of the most modular embedding dimensions for the sprites dataset, trained with an unnamed-factor oracle. See the figure caption for a description and discussion.

Figure 4: Qualitative evaluation of disentanglement on the sprites dataset. Each block of images corresponds to a single embedding dimension with very high modularity (modularity score $> 0.999$), and is marked with the attribute/factor it is associated with (pants color, body type, etc). Within a block, each row represents the approximate value of the single embedding dimension: in the first block, sprites in the top row all have gray pants and the bottom row all have yellow pants. Within a column, all factors except the factor of interest are fixed. To the extent that the images in a row share a similar factor-value, the embedding dimension has captured the factor of interest. The naive equal-width binning procedure doesn't produce perfect results — both pink and blond hair color show up in the same bin — but in general, similar factor-values are grouped together. Since the dataset does not contain all combinations of factors, some cells in the grid are missing.