[Reviews · NeurIPS 2018]

Reviewer 1



This approach tries to connect together the literature of learning deep embeddings and the literature on disentangled representation learning. In particular, it leverages the weakly supervised training process of the deep embeddings literature, which relies on the availability of data that either belongs to a class (shares a particular generative factor) or not. The authors encourage disentangling by separating the examples of the different classes based on independent per-embedding-dimension hypothesis testing using the F-statistic loss. Two examples are considered to belong to different classes based on their dissimilarity on a subset of d dimensions of the embedding, which allows the approach to be more flexible. The authors also show that this approach works when the separation is done not on class labels, but on attribute features. The paper is very well written. The approach was evaluated on a number of standard datasets and compared to state of the art baselines in terms of both few-shot classification and disentanglement. It tends to perform as well or better than the baselines on all metrics and in my opinion is worth being published at NIPS. Saying this, I have a few questions: -- Line 112. You mention sampling x_\alpha1...N and x_\beta1...M instances for each class. Shouldn't these be the same (N=M) given that it is better if N \approx M for ANOVA with unequal variances as you mention in lines 83-84? -- Line 119. In terms of the separability on d dimensions - can the model look at the top d most different dimensions per batch of a pair of classes? In other words, are these dimensions dynamically changing throughout training? -- Line 154. embeddings obtained discovered by --> embeddings discovered by Other thoughts: It would be very interesting to see how your model performs when combined with a reconstruction log likelihood loss as you suggest in line 125. In terms of the tradeoff between the modularity and explicitness of the beta-VAE, we have noticed a similar trend. In our observation it happens because the capacity of the latent bottleneck is restricted too much at high beta values. This means that not all of the data generative factors can be represented anymore. Hence, while those generative factors that are represented end up being very well disentangled, the fact that some generative factors are not encoded in the latent space at all means that the explicitness score suffers.

Reviewer 2



Summary This work seeks to combine methods from deep embedding literature with the aims of disentanglement to create disentangled embeddings. The authors use the F statistic: a well known statistical measure of inter vs intra class variance, in order to introduce separation between classes. To produce the embedding, convolutional features are used to train a fully connected embedding network with a loss designed around the F-statistic. The embeddings were evaluated against other state of the art deep metric losses such as triplet loss, structured softmax loss, and histogram loss. The F-statistic loss did not significantly outperform these other methods. The authors then evaluated the degree of disentanglement between the produced embeddings using two main metrics: modularity and explicitness. Similar to InfoGAN and Beta-VAE, dimensions of a disentangled representations are required to have mutual information with at most one factor of variation - this is defined as modularity. Explicitness is defined as the existence of a linear transformation between a dimension of the embedding, and the value of the factor of variation it represents. Here this is calculated with a one-vs-rest linear classifier, where explicitness is predicated on the ability of the classifier to predict the correct class from a single dimension of the embedding. The disentangled representations are trained in a weakly-supervised fashion, where an oracle conveys information about the similarities between batch samples, but not the name of the factor upon which they differ. The embeddings were compared against histogram loss (Ustinova & Lempitsky, 2016), triplet (Schroff et al., 2015), as well as the unsupervised Beta-VAE. Under the metrics of modularity and explicitness, the F-statistic based embeddings performed better than other methods. Review F-tests are often used to decompose the variance given in a modeled distribution. Using the F-statistic to encourage disentangled representations seems natural to me. Simply creating well structured embeddings was clearly not the point of this work, as the embeddings did not perform significantly better than other methods. However, under the two computed statistics of modularity and explicitness, the F-statistic based loss performed better than both fully supervised and unsupervised methods (even if comparing to unsupervised methods is unfair). That being said, the paper was unclear for me in a number of places: 1) I didn’t understand the training mechanism for the disentangled representations. How was 5 fold validation used here, and why was it important? A graphic showing an example curated mini-batch would help here. 2) It would be helpful to explicitly state how this measure of modularity differs from the mutual information terms which InfoGAN use in their losses to encourage disentanglement. 3) Why not directly evaluate against a method which explicitly encourages disentanglement like the F-statistic loss, such as (Chen et al. 2016) which minimizes total correlation between dimensions. 4) This could also be done in a fully supervised way, this is a stronger argument for the F-statistic in general if it is shown to beat state of the art as both a supervised and weakly-supervised method. The main idea is simple if not appreciated: using the F-statistic to encourage separation between classes. I have not seen this before in disentanglement research, but I expect to see it again in the future. I think there could have been a stronger evaluation done by comparing against other methods. If already comparing to a unsupervised method like Beta-VAE, why not compare to a method like InfoGAN which explicitly maximizes mutual information between a small subset of the code, and the observation, I believe this work was well thought-out, but it needs either a stronger evaluation, or a stronger argument for its current evaluation criterion. ** After rebuttal: I still don't like that they don't want to compare to InfoGAN. They used the metrics in the following paper (which delineates requirements for disentanglement) https://openreview.net/pdf?id=By-7dz-AZ And this paper scores InfoGAN higher than Beta-VAE on almost all metrics e.g. "InfoGAN achieves the highest average disentanglement", "InfoGAN also achieves the highest average completeness", "InfoGAN contains the most easily-extractable / explicit information about [embedding] z. So I don't buy their assertion that Beta-VAE is better than InfoGAN thus comparing to Beta-VAE is all that matters. I still believe the evaluation is insufficient because of this.

Reviewer 3



The paper proposes a loss function based on f-statistic for learning embeddings using deep neural networks and with weak supervision. The proposed loss is based on the statistics of the training datasets as compared to the common approach of using the instance level similarities/dissimilarities. Further, the class separability loss on the embedding is active only on a subset of the dimensions. The authors claim that this helps in preserving the intra-class variation information and leads to a more disentangled embeddings. Strengths: 1. The idea of using training set statistics for learning embeddings is quite interesting. 
 2. The properties of f-statistics are well described in the context of the problem addressed.
 3. Use of weak supervision for disentanglement is useful in real scenarios.
 Weakness: 1. It is not clear from limited results/experiments/datasets that the disentanglement is actually helping the few-shot learning performance the main goal of the approach.
 2. Mutual Information based metric could have some stability issues. It will be useful to discuss the limitations of the metrics and perhaps more discussion on the recent/related disentanglement metrics.
 3. In general the paper is well written, however, some information such as training details can be made clearer. 
 It will be good to clarify the details of how the relevant dimensions are chosen. This information seems to be spread between section 1.1. 2.1 and 4.1. Does the mini-batch size affect the f-statistics computation? It will also be nice to see some qualitative results for the disentanglement on the embedding. Post rebuttal comments: Thanks for the clarifications. I suggest the authors make the training details more organized so the results are easily reproduced.